# Can an E-Mail-Delivered CBT for Insomnia Validated in the West Be Effective in the East? A Randomized Controlled Trial

**DOI:** 10.3390/ijerph19010186

**Published:** 2021-12-24

**Authors:** Isa Okajima, Noriko Tanizawa, Megumi Harata, Sooyeon Suh, Chien-Ming Yang, Shirley Xin Li, Mickey T. Trockel

**Affiliations:** 1Department of Psychological Counseling, Tokyo Kasei University, Tokyo 173-8602, Japan; 2Faculty of Human Sciences, Waseda University, Saitama 359-1192, Japan; 3Department of Innovation Laboratories, NEC Solution Innovators, Ltd., Tokyo 136-8627, Japan; n-tanizawa@asagi.waseda.jp; 4Public Children Support Center at Adachi-ku, Tokyo 121-0816, Japan; speaknow64@moegi.waseda.jp; 5Department of Psychology, Sungshin Women’s University, Seoul 02844, Korea; dr.alysuh@gmail.com; 6The Research Center for Mind, Department of Psychology, Brain & Learning, National Chengchi University, Taipei 11605, Taiwan; yangcm@nccu.edu.tw; 7Department of Psychology, The University of Hong Kong, Hong Kong, China; shirleyx@hku.hk; 8The State Key Laboratory of Brain and Cognitive Sciences, The University of Hong Kong, Hong Kong, China; 9Department of Psychiatry, Stanford University, Palo Alto, CA 94305, USA; trockel@stanford.edu

**Keywords:** insomnia, depression, anxiety, self-help, cognitive behavioral therapy, self-monitoring

## Abstract

This study examined the effects of an e-mail-delivered cognitive behavioral therapy for insomnia (CBT-I), validated in Western countries, on insomnia severity, anxiety, and depression in young adults with insomnia in Eastern countries, particularly Japan. This prospective parallel-group randomized clinical trial included college students with Insomnia Severity Index (ISI) scores of ten or higher. Participants were recruited via advertising on a university campus and randomized to an e-mail-delivered CBT-I (REFRESH) or self-monitoring (SM) with sleep diaries group. The primary outcomes were insomnia severity, anxiety, and depression; secondary outcomes were sleep hygiene practices, dysfunctional beliefs, sleep reactivity, and pre-sleep arousal. All measurements were assessed before and after the intervention. A total of 48 participants (mean (SD) age, 19.56 (1.86) years; 67% female) were randomized and included in the analysis. The results of the intent-to-treat analysis showed a significant interaction effect for insomnia severity, anxiety, depression, sleep hygiene practice, and pre-sleep arousal. Compared with the SM group, the REFRESH group was more effective in reducing insomnia severity (Hedges’ g = 1.50), anxiety (g = 0.97), and depression (g = 0.61) post-intervention. These findings suggest that an e-mail-delivered CBT-I may be an effective treatment for young adults with elevated insomnia symptoms living in Japan.

## 1. Introduction

Insomnia is often persistent [1], and is a risk factor for the subsequent onset of depressive and anxiety disorders [2]. Sleep disturbances including insomnia, are prevalent among adolescents [3,4], with a prevalence rate of 22–37% in Asian countries [5,6,7]. Miyake et al. [7] reported that 36% of Japanese college students are suffering from insomnia symptoms. Research suggests that young adults with any sleep disturbances such as insomnia and sleep debt are at a higher risk for depression and suicide [8,9]. Therefore, it is important to undertake early interventions for young adults with insomnia.

Studies have revealed that cognitive-behavioral approaches for insomnia are effective in improving sleep quality and depression among college students [10,11,12]. For example, those who participated in an eight-week sleep education program showed an increase in sleep hygiene practices and improvement in sleep quality and depressive symptoms [10]. Another study by Trockel et al. [11] examined the effects of a form of cognitive-behavioral therapy for insomnia (CBT-I) delivered by e-mail, called the eight-week REFRESH program, on sleep quality measured by the Pittsburgh Sleep Quality Index (PSQI), and depressive symptoms among U.S. college students. As a results, students with poor sleep quality at baseline in the REFRESH group were associated with greater improvements in sleep quality and greater reduction in depressive symptoms than those in the stress coping control group. On the other hand, among students with high sleep quality at baseline, there was no difference in baseline compared with post-intervention changes in sleep quality or depressive symptoms.

In a systematic review and meta-analysis of self-help CBT-I for middle-aged individuals (mean age 49.3 years) [13], intervention was significantly more effective than the control group comparison in terms of insomnia, depressive and anxiety symptoms. Self-help CBT-I includes a computerized program, which has been shown to be effective in recent years [14]. However, its development has the disadvantage of being costly.

Thus, although an e-mail-delivered CBT-I may be a cost-effective intervention for students with insomnia to improve their sleep and reduce depressive symptoms, few studies have been conducted on the outcome of e-mail-delivered CBT-Is in young adults with insomnia in Asian countries. In Japan, a self-help sleep education program has been tested for teenagers [15]; however, it is not intended for young adults with insomnia, and no effective intervention has been developed for insomnia in young adults. In addition, sleep hygiene education alone has not been recommended due to a limited effect on insomnia [16].

Currently, there are 312,214 international students in Japan, and the number of Japanese students going abroad is 115,146 [17]. For bilinguals, including returnees, it is better to provide intervention programs in a language that is easy to understand and familiar to them. In Japan, there are few therapists who can provide therapy using languages other than their native tongue. If the effectiveness of this study’s e-mail-delivered intervention program, which has been validated using an English language version, is also confirmed in translated versions, it will not only ensure the effectiveness, but also the reproducibility of the REFRESH program. Therefore, this study could be a stepping stone to providing e-mail-delivered CBT-I to international students with insomnia in their native language. Furthermore, this program can be used as an evidence-based transcultural approach for cross-cultural comparative research, if this study reveals the homogeneous effect of CBT-I program.

Therefore, this study aimed to examine the effects of an e-mail-delivered CBT-I program, validated in the U.S. [11] on insomnia severity, anxiety, depression, sleep hygiene practice, and pre-sleep arousal in Japanese young adults with insomnia, compared with a control group.

## 2. Materials and Methods

This study was approved by the Ethics Committee of Waseda University (ID: 2017-191 date: 9 November 2017). All participants provided written informed consent. The study was conducted and reported per the Consolidated Standards of Reporting Trials reporting guidelines.

### 2.1. Participants

We recruited participants from 10 January to 20 December 2018, via advertising on a university campus in Japan. A total of 175 students completed the informed consent form and online questionnaires. Inclusion criteria included: (1) university students, and (2) a total Insomnia Severity Index (ISI) score ≥ 10, a clinical cut-off point to distinguish insomniac patients from normal sleepers for the Japanese version of the ISI [18]. Exclusion criteria included: (1) ISI score < 10, and (2) a current or past medical history of mental disorder, such as bipolar or schizophrenia spectrum disorder. After exclusions, the remaining 48 participants (67% female, 19.56 (SD = 1.86)) were randomly assigned to either the REFRESH group (8 males, 16 females, *n* = 24) or to a self-monitoring group (SM) that recorded their sleep diaries (8 males, 16 females, *n* = 24) for 8 weeks (Figure 1).

### 2.2. Sample Size

The sample size was based on a previous study’s power analysis conducted to assess the scores of the PSQI and Center for Epidemiologic Studies Depression Scale [11]. The effect sizes were estimated from the 8-week REFRESH program: Cohen’s d was 1.3 and 0.6, respectively, in 19 participants with poor sleep quality in the REFRESH group. With a power of 0.8, to detect a significant difference at *p* = 0.05 (2-sided), 7 to 24 individuals would be required for each group. Therefore, we recruited 24 participants in each group.

### 2.3. Study Design

This randomized clinical trial used a prospective parallel-group design. Random allocation sequences were conducted using a computer and stratified randomization by sex. Participants were randomized to the intervention (REFRESH) or self-monitoring (SM) groups. The first author enrolled and assigned participants to interventions and the second and third authors conducted weekly feedback on their sleep state, and responses to their comments via e-mails were given to the participants in both groups based on their sleep diary records. All participants were assessed for sleep-related measurements before and after the intervention. Upon completion of the study, the participants in both groups received 1000 yen per participated session.

For informed consent, the participants received information on the background and purpose of the research, type of research intervention, voluntary participation, duration, risks, benefits, confidentiality, sharing of results, right to refuse or withdraw, alternatives to participating, and whom to contact. If they agreed to participate, they were asked to sign an agreement form.

### 2.4. Measurements

#### 2.4.1. Primary Outcomes

Insomnia Severity Index (ISI). The ISI is a validated seven-item self-reported questionnaire that assesses the severity of insomnia. A summed score was calculated (range, 0–28), with higher scores indicating more insomnia symptoms [18,19]. Severity levels were categorized into: no insomnia (0–7 points), subthreshold (mild) insomnia (8–14 points), moderate insomnia (15–21 points), and severe insomnia (22 points or more). We used the Japanese version of the ISI, which was confirmed to have good reliability (α = 0.84), construct validity (r = 0.62 correlated with PSQI), and the cut-off score of the scale was 10 points in clinical and community samples [18].

Depression Anxiety Stress Scale-21 (DASS-21). The DASS is a validated 21-item self-reported scale that measures symptoms of depression, anxiety, and stress on a 4-point Likert scale [20]. The DASS-21 consists of subscales for depression, anxiety, and stress. A summed score was calculated, with higher scores indicating poorer mental health. We used the Japanese version of the DASS-21, which is available on the official website (http://www2.psy.unsw.edu.au/dass/ (accessed on 5 January 2021)).

#### 2.4.2. Secondary Outcomes

Sleep Hygiene Practice Scale (SHPS). The SHPS is a validated 30-item self-reported scale that assesses sleep hygiene practices on a 6-point Likert scale [21,22]. The SHPS consists of four domains: arousal-related behaviors, sleep scheduling and timing, eating/drinking behaviors, and sleep environment. A summed score was calculated, with higher scores indicating poorer sleep hygiene. We used the Japanese version of the SHPS [21], which was confirmed to have relatively good reliability (α = 0.54–0.74) and test-retest reliability (r = 0.55–0.76), and construct validity (r = 0.16–0.60 correlated with ISI) in the community sample.

Dysfunctional Beliefs and Attitudes about Sleep-16 (DBAS-16). The DBAS-16 scale is a validated 16-item self-reported questionnaire that assesses dysfunctional beliefs and attitudes about sleep on an 11-point Likert scale. A summed score was calculated (range, 0–160), with higher scores indicating more dysfunctional beliefs about sleep [23,24]. We used the Japanese version of DBAS [24], which was confirmed to have good reliability (α = 0.89) and construct validity (r = 0.52 correlated with ISI) in the clinical and community samples.

The Ford Insomnia Response to Stress Test (FIRST). The FIRST is a validated 9-item self-reported questionnaire that assesses sleep reactivity to stress (i.e., hyperarousal caused by a stressful event) on a 4-point Likert scale. Research has suggested that sleep reactivity is a vulnerability factor in the onset of insomnia and depression [25]. A summed score was calculated (range: 9–36), with higher scores indicating more sleep reactivity [26,27]. We used the Japanese version of FIRST [27], which was confirmed to have good reliability (α = 0.87), construct validity (r = 0.44 correlated with the State-Trait Anxiety Inventory, and r = 0.30 correlated with the Athens Insomnia Scale) in the clinical and community samples.

The Pre-Sleep Arousal Scale (PSAS). The PSAS is a validated 16-item self-reported questionnaire that assesses pre-sleep arousal on a 4-point Likert scale. The PSAS was constructed with two-factor somatic (i.e., physiological) arousal and cognitive arousal subscales [28,29]. The score for each subscale was summed (range: 8–40), with higher scores indicating a state of greater pre-sleep arousal. We used the Japanese version of PSAS [29], which was confirmed to have relatively good reliability (α = 0.85–0.90) and test-retest reliability (r = 0.67–0.78), and construct validity (r = 0.43–0.52 correlated with ISI, r = 0.38–0.53 correlated with FIRST, and r = 0.35–0.44 correlated with DBAS) in the community sample.

### 2.5. Intervention

#### 2.5.1. Intervention (REFRESH) Group

The intervention was delivered in eight weekly sessions via e-mail messages with attached PDF files. Students were encouraged to spend 30 min for each session. We obtained permission from the first and corresponding authors of the original program [11] to translate the original program from English into Japanese.

The REFRESH program consists of (1) the physiology of sleep, with particular emphasis on circadian rhythms and recommendations for stabilizing the circadian rhythm through anchoring wake time; (2) instructions on a time-in-bed restriction protocol to consolidate sleep; (3) relaxation training; (4) mindfulness training; (5) stimulus control strategies; and (6) cognitive strategies to reduce the impact of maladaptive thoughts about sleep [11]. The program encouraged participants to keep daily sleep diaries and implement strategies to improve sleep health. In each session, homework was assigned and participants were encouraged to engage in the homework for at least 30 min. Due to e-mail delivery, information about the time spent on each session module could not be collected.

The daily sleep diaries online allowed the participants to record daily bedtimes, time out of bed, sleep onset latency, wake after sleep onset, number of nocturnal awakenings, total hours of sleep, number of alcoholic beverages before bedtime, satisfaction with sleep, degree of feeling refreshed in the morning, and individual comments. Participants were asked to record their sleep diary immediately after waking up. The participants in the REFRESH group received weekly feedback on their sleep state and responses to their comments based on their sleep diary via e-mail messages.

#### 2.5.2. Self-Monitoring Group

As with the REFRESH group, the participants in the SM group maintained their daily sleep diaries and implemented strategies for improving sleep health for eight weeks. Research has shown SM using an online sleep diary to be more effective in improving insomnia than a wait-list control group [30]. Participants were asked to record their sleep diary immediately after waking up. Weekly feedback on their sleep state and responses to their comments via e-mail messages were given to the participants in the SM group based on their sleep diary records.

### 2.6. Statistical Analysis

All data were analyzed using SPSS (version 25.0; IBM Inc., Tokyo, Japan) and R statistical software (version 3.6.3; R Project for Statistical Computing, Vienna, Austria). To examine the effect of the REFRESH program on insomnia-related symptoms and depressive symptoms, a mixed-effect model for repeated measures (MMRM), which does not compensate for missing data, was used to compare the data within and between groups.

Additionally, we estimated the effect sizes of the scales within and between groups by correcting biases for Hedges’ g. In general, an absolute g value of 0.2 or more indicates a small effect size; approximately 0.5, moderate; and 0.8 or more, large [31]. The effect sizes of all scales between groups were analyzed post-intervention. For all scales, the more positive the change in the effect size, the larger its therapeutic effect.

To confirm the clinically significant improvement of the REFRESH program, we calculated the number needed to treat (NNT) using the remission rate of insomnia for each group. As for the remission rate, ISI < 8 points [19] was used so that the results could be compared with those of previous studies of CBT-I.

Harmful events would be judged if the participants reported a significant worsening of mental conditions (e.g., onset of manic or depressive symptom) during the start of the intervention.

## 3. Results

### 3.1. Adherence and Attrition

Of the 48 participants, one dropped out of the SM group and three in each group did not complete the post-intervention questionnaires. The completion rates of each intervention were 88% in the REFRESH group and 83% in the SM group. Adherence was measured by using the number of recording sleep diaries per week for each group. The weekly average number of recording sleep diaries was 5.92 (SE = 0.44) in the REFRESH group and 5.89 (SE = 0.44) in the SM group. In total, 95% of the REFRESH group and 90% of the SM group reported completing ≥4 of 8 sessions. Of them, 90% and 80% completed ≥7 sessions.

No harmful events were reported. At pre-intervention, there were no significant differences among the groups in any outcome measures. Table 1 lists the descriptive statistics of all measures for each group pre- and post-intervention.

### 3.2. Main Outcomes: Insomnia Severity, Depression, Anxiety, and Stress

The results showed significant effects of group (F_1,46_ = 14.19, *p* < 0.001), time (F_1,44_ = 46.89, *p* < 0.001), and group by time interaction (F_1,44_ = 16.46, *p* < 0.001) on the ISI score. In the results of the post-hoc test, the REFRESH group showed a significantly greater improvement in insomnia symptoms at the post-intervention period when compared with the SM group (*p* < 0.001; Hedges’ g = 1.50 (95% CI, 0.85 to 2.15); Figure 2a). Additionally, the ISI score significantly improved from pre- to post-intervention in the REFRESH group (*p* < 0.001).

For depression, there was a significant effect of time (F_1,43_ = 10.70, *p* = 0.002) and group by time interaction (F_1,43_ = 4.43, *p* = 0.041) on the DASS-depression subscale score. In the results of the post-hoc test, the REFRESH group had significantly improved depressive symptoms at post-intervention compared with the SM group (*p* = 0.035; g = 0.61 (0.02 to 1.19); Figure 2a). The DASS-depression score significantly improved from pre- to post-intervention in the REFRESH group (*p* < 0.001).

For anxiety, there was a significant effect of group (F_1,46_ = 7.33, *p* = 0.009), time (F_1,42_ = 13.12, *p* = 0.001), and group by time interaction (F_1,42_ = 5.35, *p* = 0.026) on the DASS-anxiety subscale score. In the results of the post-hoc test, the REFRESH group showed significantly greater improvement in anxiety symptoms at post-intervention compared with the SM group (*p* = 0.001; g = 0.97 (0.36 to 1.57); Figure 2a). Further, the DASS-anxiety score significantly improved from pre- to post-intervention in the REFRESH group (*p* < 0.001).

For stress, there was a significant effect of group (F_1,48_ = 8.49, *p* = 0.001) and time (F_1,44_ = 20.13, *p* < 0.001) but no significant group by time interaction (F_1,44_ = 3.13, *p* = 0.084) for the DASS-stress subscale score. The effect size was large (g = 0.93 (0.33 to 1.54); Figure 2a).

### 3.3. Secondary Outcomes: Sleep Hygiene Practice, Stress Reactivity, Dysfunctional Beliefs, and Pre-Sleep Arousal

For sleep hygiene practice, there was a significant group by time interaction effect on subscale scores for sleep scheduling and timing (F_1,42_ = 8.56, *p* = 0.006), arousal-related behaviors (F_1,42_ = 5.47, *p* = 0.024), and eating/drinking behaviors (F_1,43_ = 5.88, *p* = 0.019). In the results of the post-hoc test, the REFRESH group had significantly greater improvement in sleep hygiene practice scores at post-intervention compared with the SM group (*p* < 0.05; g = 1.03 (0.42 to 1.64) for sleep scheduling and timing; g = 1.00 (0.39 to 1.61) for arousal-related behaviors; and g = 0.58 (0.00 to 1.16) for eating/drinking behaviors; Figure 2b). In addition, the scores for each subscale significantly improved from pre- to post-intervention in the REFRESH group (*p* < 0.01). The sleep environment subscale did not show a significant group by time interaction effect, and the effect size was small (g = 0.29 (−0.29 to 0.86)).

For stress reactivity, there was a significant effect of time (F_1,39_ = 12.76, *p* < 0.001) but no significant group by time interaction (F_1,44_ = 3.13, *p* = 0.084) for the FIRST score. The effect size between groups was small (g = 0.24 (−0.34 to 0.81); Figure 2b); however, the score changes from pre- to post-intervention had a moderate effect (g = 0.71 (0.12 to 1.30)).

There was a significant effect of time (F_1,43_ = 20.97, *p* < 0.001) but no significant group by time interaction (F_1,43_ = 3.13, *p* = 0.082) for the DBAS score. The effect size between groups was moderate (g = 0.60 (0.01 to 1.18); Figure 2b), but the pre- to post-intervention score change had a large effect (g = 1.02 (0.41 to 1.63)).

Pre-sleep arousal showed significant group by time interaction effects on the subscale scores of cognitive arousal (F_1,42_ = 5.68, *p* = 0.022) and somatic arousal (F_1,42_ = 7.61, *p* = 0.009). In the results of the post-hoc test, the REFRESH group had significantly greater improvement in both cognitive and somatic arousal at post-intervention when compared with the SM group (*p* < 0.05; g = 0.92 (0.32 to 1.52) for cognitive arousal; g = 1.04 (0.43 to 1.65) for somatic arousal; Figure 2b). In addition, the scores for each subscale significantly improved from pre- to post-intervention in the REFRESH group (*p* < 0.001).

### 3.4. Clinically Significant Improvement

The REFRESH group had a 52% remission rate (ISI < 8) of insomnia and the SM group had a 0% remission rate (NNT = 1.9 (1.4 to 3.2)) at post-intervention.

## 4. Discussion

This study aimed to examine the effects of an e-mail-delivered CBT-I (REFRESH) on insomnia severity, affective symptoms such as depression, and insomnia-related symptoms in young adults with insomnia, compared with a SM group. As well as the original program [11], the results showed that the program is more effective in improving insomnia and depression, and additionally anxiety, and in reducing poor sleep hygiene and pre-sleep arousal, than SM. The findings suggest an e-mail-delivered CBT-I, validated previously using an English language version in the West, could be effective with a version translated into participants’ native language in the East.

### 4.1. Effects in Improvement of Insomnia and Affective Symptoms

The results showed that an e-mail-delivered CBT-I is effective in improving the symptoms of insomnia, depression, and anxiety, compared with SM. In addition, it showed that the improvement in insomnia severity was large (Hedges’ g = 1.50), and the clinically significant improvement was encouraging (NNT = 1.9). The findings are in line with previous studies, which conducted a face-to-face CBT-I for adolescents with insomnia comorbid psychiatric disorders and chronic pain [32]. The study showed that statistically significant improvements were found for insomnia symptoms (d = 1.63), depression (d = 0.87), and anxiety (d = 0.31).

In a systematic review and meta-analysis of self-help CBT-I for middle-aged individuals [13], intervention was significantly more effective than the control group comparison in terms of sleep questionnaire scores (g = 1.26) [13]. Intervention was also significantly better than the control group comparison in reducing depressive (g = 0.50) and anxiety symptoms (g = 0.42) [13]. Results of the current study add to extant evidence that amelioration of insomnia may decrease anxiety and depressive symptoms [33].

Taken together, results of the current study coupled with previous evidence suggest an e-mail-delivered CBT-I program can be effective for improving insomnia, depression, and anxiety symptoms in young adults with insomnia symptoms at baseline.

### 4.2. Reduction in Insomnia-Related Symptoms

The results showed that the REFRESH program was effective in improving sleep hygiene practices and pre-sleep arousal when compared with SM.

Intervention effects on arousal-related behaviors (e.g., checking the time in the middle of the night), sleep scheduling and timing (e.g., inconsistent daily bedtimes), and eating/drinking behaviors (e.g., drinking caffeinated drinks four hours prior to bedtime) were moderate to large. These findings are consistent with a previous study in which sleep-related behaviors perpetuated insomnia [34]. In contrast, research has shown the effectiveness of sleep hygiene education components of CBT-I to be of limited utility for patients [16]. The current study’s results suggest that a CBT-I program that includes several components including sleep hygiene education may be an effective tool to improve sleep hygiene behavior in young adults with insomnia.

The effect sizes for both cognitive and somatic arousal were large. These results are in line with the findings of a previous study, in which CBT-I was found to reduce pre-sleep arousal in insomnia patients [35]. However, there were improvements in scores for sleep reactivity and dysfunctional beliefs about sleep in both groups post-intervention and there were no significant differences between groups in either of these two parameters. In these scales, there were moderate to large changes in scores from pre- to post-intervention for sleep reactivity (g = 0.71) and dysfunctional beliefs about sleep (g = 1.02). Researchers have suggested that hyperarousal may be a key cause of insomnia [36,37]. A recent study has suggested that hyperarousal includes a trait predisposition toward excess arousal (e.g., sleep reactivity) and pre-sleep-state-dependent hyperarousal (e.g., pre-sleep arousal) [36]. Studies have also reported that CBT-I improves trait and state arousal and dysfunctional beliefs [33,35].

Thus, an e-mail-delivered CBT-I is effective in improving insomnia in young adults with elevated insomnia symptoms, but only 52% of them showed benefit. According to the stepped-care model of insomnia [38], it may be necessary to offer a face-to-face CBT-I.

### 4.3. Limitation

This study has some limitations. First, since we used the cut-off score of the ISI, it is possible that the study participants included sleep disturbances other than insomnia (e.g., delayed sleep-wake phase disorder), although individuals with a current or past medical history of mental disorders were excluded. In the future, it will be necessary to assess subjects strictly using the diagnostic criteria of sleep disorders. Second, the participants of this study were paid a fee for participation. If no honorarium was paid, the completion rate of a self-help program has been reported to be 63% [10]. Therefore, the high rate of adherence and attrition might have been due to this fact. Although the sample size was based on a previous study’s power analysis, the results of this study may have been biased, because of the reduced sample size due to dropout. Third, we could not collect information about time spent on each session module. Fourth, objective measurements of sleep, such as actigraphy, were not included in this study. Finally, since there was no set follow-up period, it is unclear whether the effects of an e-mail-delivered CBT-I will be sustained.

## 5. Conclusions

To the best of our knowledge, this is the first study on the effect of an e-mail-delivered CBT-I on insomnia severity, anxiety, and depression in young adults in Japan. The findings are notable in that they suggest that the REFRESH program, which has been validated previously using an English language version, can be effective for young adults living in Asian countries when translated into the native language of participants. Furthermore, it could be able to contribute as an evidence-based transcultural approach for cross-cultural comparative study.

For students who complain of insomnia symptoms and find face-to-face CBT-I difficult or refuse it, an e-mail-based program can be suggested. For international students and returnees, we can offer a language that is easy for them to understand. Since conducting a face-to-face CBT-I would have been difficult during the coronavirus (COVID-19) pandemic, it is noteworthy that this study demonstrated the effectiveness of e-mail-delivered CBT-I on symptoms related to insomnia, depression, and anxiety. In future research, it is necessary to examine whether the program is also effective for COVID-19-related insomnia, depression, and anxiety. In addition, since the effectiveness of fully automated digital CBT-I has been currently clarified [14], it is necessary to confirm whether a digital-CBT-I is effective as a transcultural approach in future research.

## Figures and Tables

**Figure 1 ijerph-19-00186-f001:**
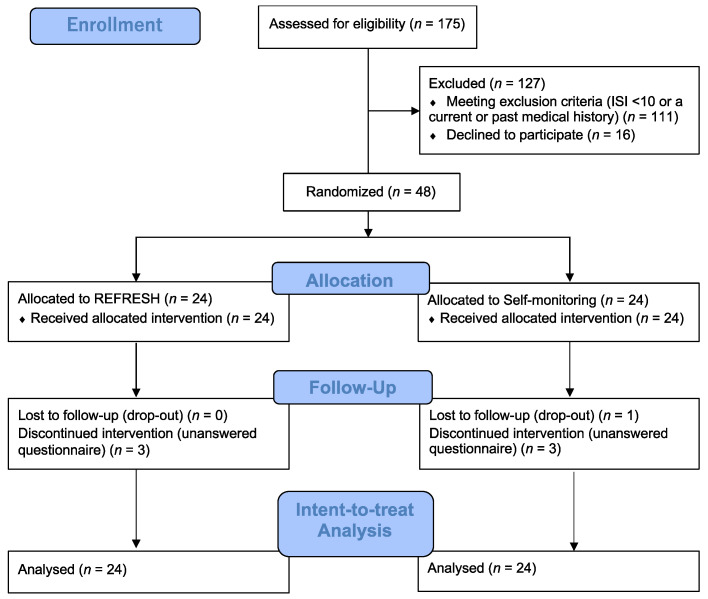
Study flowchart.

**Figure 2 ijerph-19-00186-f002:**
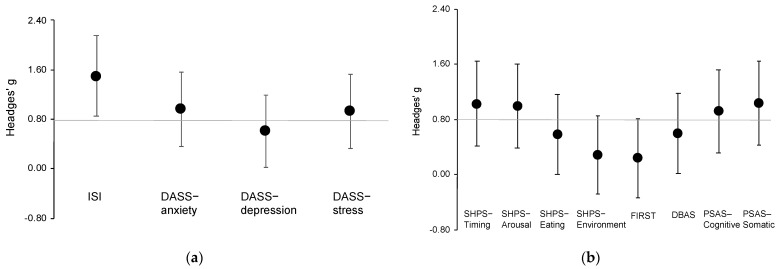
Plots of effect sizes through time for outcome measures between groups. (**a**) Effect sizes for main outcomes; (**b**) effect sizes for secondary outcomes. DASS, Depression Anxiety Stress Scale; DBAS, Dysfunctional Beliefs and Attitudes about Sleep; Error bars indicate 95% CI; FIRST, Ford Insomnia Response to Stress Test; ISI, Insomnia Severity Index; PSAS, Pre-Sleep Arousal Scale; SHPS, Sleep Hygiene Practice Scale. The horizontal line indicates a large effect size (g = 0.8).

**Table 1 ijerph-19-00186-t001:** Outcome measures in each group.

Scales	REFRESH Group		Self-Monitoring Group		*p*-Values		
	Pre	Post	Pre	Post	Group	Time	Interaction
ISI	13.63 (0.66)	7.01 (0.70)	14.00 (0.66)	12.31 (0.72)	**<0.001**	**<0.001**	**<0.001**
DASS-anxiety	3.97 (0.53)	1.55 (0.56)	4.79 (0.53)	4.27 (0.57)	**0.009**	**0.001**	**0.026**
DASS-depression	6.29 (0.81)	2.97 (0.85)	6.29 (0.81)	5.57 (0.87)	ns	**0.002**	**0.041**
DASS-stress	6.83 (0.73)	3.10 (0.78)	8.38 (0.73)	6.75 (0.79)	**0.005**	**<0.001**	0.084
SHPS-timing	26.79 (1.29)	19.87 (1.36)	28.21 (1.29)	26.91 (1.38)	**0.012**	**<0.001**	**0.006**
SHPS-arousal	28.83 (1.33)	22.29 (1.42)	29.75 (1.33)	29.43 (1.45)	**0.007**	**0.014**	**0.024**
SHPS-eating	16.04 (0.77)	12.99 (0.78)	14.96 (0.77)	15.39 (0.84)	ns	0.074	**0.019**
SHPS-environment	22.68 (1.25)	21.24 (1.32)	23.96 (1.25)	23.15 (1.35)	ns	ns	ns
FIRST	25.46 (1.00)	22.58 (1.05)	26.13 (1.00)	23.83 (1.07)	ns	**0.001**	ns
DBAS	91.96 (3.31)	74.89 (3.50)	92.88 (3.31)	85.37 (3.57)	ns	**<0.001**	0.082
PSAS-cognitive	24.67 (1.28)	16.99 (1.36)	25.54 (1.28)	23.28 (1.39)	**0.021**	**<0.001**	**0.022**
PSAS-somatic	16.33 (1.05)	12.43 (1.09)	18.17 (1.05)	18.13 (1.11)	**0.007**	**0.007**	**0.009**

DASS, Depression Anxiety Stress Scale; DBAS, Dysfunctional Beliefs and Attitudes about Sleep; FIRST, Ford Insomnia Response to Stress Test; ISI, Insomnia Severity Index; ns, not significant; PSAS, Pre-Sleep Arousal Scale; SHPS, Sleep Hygiene Practice Scale. Standard Errors are shown in parentheses. Bold indicates that the *p*-value is less than 0.05.

## Data Availability

The datasets analyzed in the current study and the Japanese version of the REFRESH program are available from the corresponding author upon reasonable request.

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
