# Peer review of "Can an E-Mail-Delivered CBT for Insomnia Validated in the West Be Effective in the East? A Randomized Controlled Trial"

_ijerph, 2021, doi:10.3390/ijerph19010186_

Round 1
Reviewer 1 Report
Thank you for asking me to review this paper. I have several suggestions for the developments.
1) Line 35-36, it is important to remember that a standard paragraph should have at least three to four lines. Please edit your paragraph.
2) Line 44-51, is it only support by one study? Are there any additional studies as your reference?
3) For the Introduction and Literature Review sections, can the researchers add some more additional references and literature to support your ideas? Currently, From Reference 1 to 14, there are only a few focused on the situation in Japan. As this study is focusing on the Japanese university environment, it is important to add some studies with the East Asian environment/Japanese environment.
3-1) For reference #8, what is the journal of this article? Is Null a journal?
4) What about the research gaps? The reviewer understands there are some problems for sleepless for Japanese students.
4-1) However, why this study is important for the field?
4-2) What are the research gaps?
4-3) How can this study fill the gaps in the field?
5) Line 64, the researchers mentioned there are 312,214 international students in Japan, which is good. However, why Japanese students are selected? Are they unique? The researchers need to explain why this study select Japan as the site (why not South Korea)?
6) For Section 4.3. Limitation, as this study was completed nearly three years ago, the results of this study may be different due to the current global health crisis. The authors should mention the current situations can be different due to the COVID-19 pandemic.
7) For Chapter 5, the researchers should add some ideas about "how can this study contribute to the practice". The study is very interesting for sure. But how can readers, counsellors, teachers, public health professionals use the results and data of this study to help their patients and students? The reviewer read the article. But the reviewer also wants to use the results to help the students. But the articles did not provide some practices and ways.
8) If the researchers can provide the surveys/interview questions/questions that were asked during the data collection procedures as the appendix, that will definitely help other readers and teachers to apply the results of the study to their university environment.
Author Response
We would like to thank you for the time you spent reviewing this manuscript. We have found your comments very useful; they have helped us to improve the manuscript significantly. Listed below are the comments you provided and explanations regarding how we have addressed them.
1). Line 35-36, it is important to remember that a standard paragraph should have at least three to four lines. Please edit your paragraph.
<Reply> We have revised the manuscript to address this comment.
2) Line 44-51, is it only support by one study? Are there any additional studies as your reference?
<Reply> We have revised the manuscript to address this comment.
Revised manuscript; Line 42 to 46:
Studies have revealed that cognitive-behavioral approaches for insomnia are effective in improving sleep quality and depression among college students [10–12]. For example, those who participated in an eight-week sleep education program showed an increase in sleep hygiene practices and improvement in sleep quality and depressive symptoms [10]. Another study showed that Trockel et al. [11] examined…
3) For the Introduction and Literature Review sections, can the researchers add some more additional references and literature to support your ideas? Currently, From Reference 1 to 14, there are only a few focused on the situation in Japan. As this study is focusing on the Japanese university environment, it is important to add some studies with the East Asian environment/Japanese environment.
<Reply> Thank you for useful comment. We have added to ref.7 and 12, and revised the manuscript to address this comment.
Revised manuscript; Line 38 to 39:
Miyake et al. [7] has been reported 36% of Japanese college students are suffering from insomnia symptoms.
3-1) For reference #8, what is the journal of this article? Is Null a journal?
<Reply> We have revised the manuscript to address this comment.
Revised manuscript; Ref.9:
Williams, A.B.; Dzierzewski, J.M.; Griffin, S.C.; Lind, M.J.; Dick, D.; Rybarczyk, B.D. Insomnia Disorder and Behaviorally Induced Insufficient Sleep Syndrome: Prevalence and Relationship to Depression in College Students. Behavioral Sleep Medicine 2020, 18, 275–286, doi:10.1080/15402002.2019.1578772.
4) What about the research gaps? The reviewer understands there are some problems for sleepless for Japanese students.
4-1) However, why this study is important for the field?
4-2) What are the research gaps?
4-3) How can this study fill the gaps in the field?
<Reply> Thank you for useful comment. The research gap of this study is to examine whether a self-help intervention program is effective for Japanese college students suffering from insomnia. There is no effective approach for them. In addition, if the effects of the translated CBT-I program are found to be homogeneous, it can be used as an approach for evidence-based cross-cultural comparative research. However, there are no studies that have examined these issues. Although sleep loss is a serious problem among Japanese, symptoms of sleep loss and insomnia are different, and therefore, different approaches are expected. This study did not focus on sleep loss because the approach of this study was conducted for insomnia symptoms. We have added the sentences as follows:
Revised manuscript; Line 64 to 65:
however, it is not intended for young adults with insomnia, and no effective intervention has been developed for insomnia in young adults.
Revised manuscript; Line 69 to 72:
For bilinguals, including returnees, it is better to provide intervention programs in a language that is easy to understand and familiar to them. In Japan, there are few therapists who can provide therapy using languages other than their native tongue.
5) Line 64, the researchers mentioned there are 312,214 international students in Japan, which is good. However, why Japanese students are selected? Are they unique? The researchers need to explain why this study select Japan as the site (why not South Korea)?
<Reply> This research is an international collaboration, and the REFRESH program had been also conducted in South Korea. This paper is a research program planned in Japan.
6) For Section 4.3. Limitation, as this study was completed nearly three years ago, the results of this study may be different due to the current global health crisis. The authors should mention the current situations can be different due to the COVID-19 pandemic.
<Reply> Thank you for useful comment. We have revised the manuscript to address this comment.
Revised manuscript; Line 378 to 380:
In future research, it is necessary to examine whether the program is also effective for COVID-19-related insomnia, depression, and anxiety.
7) For Chapter 5, the researchers should add some ideas about "how can this study contribute to the practice". The study is very interesting for sure. But how can readers, counsellors, teachers, public health professionals use the results and data of this study to help their patients and students? The reviewer read the article. But the reviewer also wants to use the results to help the students. But the articles did not provide some practices and ways.
<Reply> Thank you for useful comment. We have revised the manuscript to address this comment.
Revised manuscript; Line 373 to 375:
For students who complain of insomnia symptoms and find face-to-face CBT-I difficult or refuse it, an e-mail-based program can be suggested. For international students and returnees, we can offer a language that is easy to understand for them.
8) If the researchers can provide the surveys/interview questions/questions that were asked during the data collection procedures as the appendix, that will definitely help other readers and teachers to apply the results of the study to their university environment.
<Reply> Feedback during each session was given on the results of each participants' sleep diary. Their free to comments in their sleep diaries were included in personal events. Therefore, it is difficult to publish them as an appendix. In addition, I did not take the post-questionnaire. I added the following sentence to Data Availability Statement section;
Revised manuscript; Line 395 to 396:
The datasets analyzed in the current study and the Japanese version of REFRESH program are available from the corresponding author upon reasonable request.
Reviewer 2 Report
Well-written paper.
Please consider the following suggestions and questions:
section Participants:
excluded (127) meeting exclusion criteria (ISI<10) (n=111) and decline to participate (n=16) it means that nobody meets second criteria of exclusion "a current or past medical history of mental disorder, such as bipolar or schizophrenia spectrum disorder". Is that correct?
We have information about the numbers of males/females in the group of 48 participants but not in the FRESH and the self-monitoring group.
section Intervention:
Did you monitor the time which students spend for each session?
If not the information should be in this section?
What was the form of a sleep diary? Is there an electronic version? Did you have control over when participants put data into the diary?
How often did participants send sleep diaries to you?
Author Response
We would like to thank you for the time you spent reviewing this manuscript. We have found your comments very useful; they have helped us to improve the manuscript significantly. Listed below are the comments you provided and explanations regarding how we have addressed them.
- section Participants:
excluded (127) meeting exclusion criteria (ISI<10) (n=111) and decline to participate (n=16) it means that nobody meets second criteria of exclusion "a current or past medical history of mental disorder, such as bipolar or schizophrenia spectrum disorder". Is that correct?
<Reply> Thank you for useful comment. We have revised the Figure 1 to address this comment.
- We have information about the numbers of males/females in the group of 48 participants but not in the FRESH and the self-monitoring group.
<Reply> We have revised the manuscript to address this comment.
Revised manuscript; Line 96 to 99:
After exclusions, the remaining 48 participants (67% female, 19.56 [SD = 1.86]) were randomly assigned to either the REFRESH group (8 males, 16 females, n = 24) or to a self-monitoring group (SM) that recorded their sleep diaries (8 males, 16 females, n = 24) for 8 weeks (Figure 1).
- section Intervention:
Did you monitor the time which students spend for each session?
If not the information should be in this section?
<Reply> We have revised the manuscript to address this comment.
Revised manuscript; Line 183 to 184:
Due to e-mail delivery, information about time spent on each session module could not be collected.
- What was the form of a sleep diary? Is there an electronic version? Did you have control over when participants put data into the diary?
<Reply> We have revised the manuscript to address this comment.
Revised manuscript; Line 185 to 189:
The daily sleep diaries via online allowed the participants to record daily bedtimes, time out of bed, sleep onset latency, wake after sleep onset, number of nocturnal awakenings, total hours of sleep, number of alcoholic beverages before bedtime, satisfaction with sleep, degree of feeling refreshed in the morning, and individual comments. Participants were asked to record their sleep diary immediately after waking up.
Revised manuscript; Line 194 to 197:
Research has shown SM using a sleep diary via online to be more effective in improving insomnia than a wait-list control group [30]. Participants were asked to record their sleep diary immediately after waking up.
- How often did participants send sleep diaries to you?
<Reply> Thank you for useful comment. We have corrected the sentence to address this comment.
Revised manuscript; Line 223 to 224:
Weekly average number of recording sleep diaries were 5.92 (SE=0.44) in the REFRESH group and 5.89 (SE=0.44) in the SM group.
Reviewer 3 Report
Interesting work performed by authors who investigated an e-mail-delivered CBT intervention. It is an original approach with the current world where telemedicine is growing faster. Despite the small sample size, the topic is original and easy to read.
Author Response
We would like to thank you for the time you spent reviewing this manuscript. We have found your comments very useful. Thank you very much.